# Quantitative Characterization of the Impact of Protein–Protein Interactions on Ligand–Protein Binding: A Multi-Chain Dynamics Perturbation Analysis Method

**DOI:** 10.3390/ijms25179172

**Published:** 2024-08-23

**Authors:** Lu Li, Hao Li, Ting Su, Dengming Ming

**Affiliations:** College of Biotechnology and Pharmaceutical Engineering, Nanjing Tech University, 30 South Puzhu Road, Jiangbei New District, Nanjing 211816, China202261118018@njtech.edu.cn (T.S.)

**Keywords:** protein–protein interaction, ligand-binding region, LPIs, PPIs, *mc*DPA

## Abstract

Many protein–protein interactions (PPIs) affect the ways in which small molecules bind to their constituent proteins, which can impact drug efficacy and regulatory mechanisms. While recent advances have improved our ability to independently predict both PPIs and ligand–protein interactions (LPIs), a comprehensive understanding of how PPIs affect LPIs is still lacking. Here, we examined 63 pairs of ligand–protein complexes in a benchmark dataset for protein–protein docking studies and quantified six typical effects of PPIs on LPIs. A multi-chain dynamics perturbation analysis method, called *mc*DPA, was developed to model these effects and used to predict small-molecule binding regions in protein–protein complexes. Our results illustrated that the *mc*DPA can capture the impact of PPI on LPI to varying degrees, with six similar changes in its predicted ligand-binding region. The calculations showed that 52% of the examined complexes had prediction accuracy at or above 50%, and 55% of the predictions had a recall of not less than 50%. When applied to 33 FDA-approved protein–protein-complex-targeting drugs, these numbers improved to 60% and 57% for the same accuracy and recall rates, respectively. The method developed in this study may help to design drug–target interactions in complex environments, such as in the case of protein–protein interactions.

## 1. Introduction

The interactions of proteins with other molecules are the basis of many essential physiological processes, such as the cytoskeleton assembly, muscle energy supply, enzymatic reactions, cell signaling, etc. [1,2]. These molecules can be biological macromolecules, such as proteins, peptides, nucleic acids, and membranes, or metabolic small molecules, substrates, and signaling small molecules, such as solvents, metal non-metal ions, and organic small molecules [3,4]. It is commonly accepted that protein–protein interactions (PPIs) create a large and relatively flat unique contact surface, whereas ligand–protein interactions (LPIs) create a rather deep, narrow, and distinct region of action. Thus, these two types of interactions may tend to exist independently of each other. However, in recent years, it has been increasingly recognized that ligands are involved in the PPI process, and the study of the interaction between LPIs and PPIs is essential to fully understand many complex cellular processes [5,6].

It was recently discovered that LPIs can be used to modulate the PPI action network to fight against disease [7]. Many small-molecule inhibitors have been developed to modulate PPI by regulating the binding sites of small molecules to proteins, and some of them have already entered the clinical treatment phase [6]. The successful development of these small-molecule inhibitors suggests that there may be a strong link between small-molecule binding pockets and PPIs. Statistics show that during its life cycle, a protein can successively make contact with multiple chaperone molecules, each of which may bind to a different site on the surface of that protein, or the protein surface may present a shared binding region that different chaperones will use at various moments of its life cycle, of which up to 75% may be used for PPIs [8]. Furthermore, Skolnick and colleagues statistically found that two out of three PPI interfaces contain at least one significant small-molecule ligand-binding pocket, suggesting that ligand-binding pockets play an important role in PPIs [5]. Two different ligands can sometimes interact with different structures in the same type of pocket in a protein. For example, the ATP-binding pocket of protein kinase p38α can accommodate two drugs, imatinib [9] and sorafenib [10], which both inhibit ATP binding in the same pocket. This feature of the small-molecule binding pockets of regulating protein–protein interactions has led to the design of many novel inhibitors [11].

The direct physicochemical interplay between PPIs and ligand-binding pockets were also observed. Bessman and colleagues [12] found that competition exists between the epidermal growth factor receptor (EGFR) dimerization (PPI) and its binding to the ligand of EGF (LPI). In this case, ligand binding promotes dimerization, but dimerization does not optimize the ligand binding. Gao and colleagues [13] demonstrated that a new ligand-binding pocket can be formed close to the protein–protein interface when two proteins interact, and this pocket may gradually vanish when the two proteins are pulled apart. Indeed, small-molecule ligands are known for their great conformational diversity. This diversity can maximize the complementarity of ligands with different surface pockets, including new surface pockets that form due to protein–protein interactions [14,15]. Moreover, it has long been recognized that protein dynamic conformational distribution also plays an important role in determining the selectivity and specificity of bound small molecules. Therefore, the effect of PPI-induced changes in protein dynamics on the binding of small-molecule ligands to proteins should not be ignored [16].

It is therefore of general interest to study how PPIs alter the ways in which ligands bind to proteins, especially for alterations in the binding pocket. Although methods have been developed to predict the protein–small-molecule binding pocket and the protein–protein interaction interface, a quantitative characterization of the impact of PPIs on LPIs is still lacking. Here, we first systematically characterized the typical changes in the small-molecule ligand-binding pocket induced by PPIs. We then predicted these changes by constructing a multi-chain dynamic perturbation analysis model (*mc*DPA). The model is based on a previously developed protein dynamics perturbation analysis (DPA) method [17]. DPA was originally designed to identify ligand-binding pockets in single-chain proteins, particularly in single structural domains. In the *mc*DPA, interactions between proteins in multi-chain systems were introduced to quantitatively characterize the alteration of small-molecule binding pockets caused by PPIs. We used a benchmark dataset of complex structures for protein–protein docking studies, from which we selected complex structures with small molecules bound to them. We then compared the changes in the small-molecule binding pocket before and after adding protein–protein interactions. The statistical results showed that upon binding to the second protein, the small-molecule binding pocket of the original protein undergoes various changes, such as retention, splitting, merging, and the creation of new pockets. The calculations also showed that the *mc*DPA was able to accurately model the effect of the newly added protein on the small-molecule binding pocket of the original protein. This study provides a better understanding of the interconnection between PPIs and LPIs, which might help in the design and development of small-molecule drugs by modulating the PPI network.

## 2. Results and Discussion

### 2.1. Protein–Protein Interactions Have Diverse Effects on the Binding of Ligands to Proteins

In the present study, we performed structural and compositional comparisons of cases where the same protein bound the same ligand in two different states: monomeric and complexed with other proteins. Through this process, we obtained a total of 63 valid pairs of data from a standard benchmark dataset for protein docking (ZDOCK) studies. Table 1 lists the ligand atomic root mean square deviation (ARMSD), which represents the conformational changes in the ligand from binding a monomeric protein to binding the corresponding multichain protein complex. Figure 1 summarizes the six typical changes in ligand–protein binding following the addition of PPI interactions.

In the first case, the binding pocket and ligand conformation remain almost unchanged, with an ARMSD value typically less than 1 Å (Figure 1A). An example of this case is in the OX category, where one structure is the GDP-bound tubulin-like protein, FtsZ, from *P. aeruginosa* (PDB ID: 2VAW [18]). The other example is the complex formed between an SOS cell division inhibitor protein, SulA, and the GDP-bound FtsZ (PDB ID: 1OFU [19]). The calculated ARMSD value for the two ligands, GDP, is small, at only 0.83 Å, indicating that the protein configuration of FtsZ and its ligand-binding conformation are not significantly affected by SulA binding. SulA inhibits FtsZ polymerization not by blocking the nucleotide-binding site where GDP binds, but by binding to the T7 loop surface of FtsZ, which is opposite the nucleotide-binding site. Nonetheless, the PPI between the SulA and the FtsZ still caused observable perturbation in the GDP-FtsZ binding. Of the sixteen binding sites, one binding site, M105, disappears, and a new binding site, N166, emerges.

The second case is characterized by a shift in the ligand-binding position or a distortion of the pocket and a significant change in the ligand conformation, resulting in a larger ARMSD (usually greater than 1 Å, Figure 1B). An example of this case is in the OG category, where one structure is the human Rac protein (PDB ID: 1MH1 [20]), and the other is a complex between the Rac and the GTP activating protein (GAP) domain of the toxin, ExoS, from *P. aeruginosa* (PDB ID: 1HE1 [21]). The ARMSD is 1.76Å for the two binding nucleotide ligands, GDP and GTP analog (the outermost phosphate group of the GTP was not counted in the ARMSD compared to the GDP). Introducing the toxin, ExoS, causes the disappearance of the original binding sites, G12, F28, E31, P34, T35, and G60, leading to the new sites, R146, T186, T187, and G188.

In the third case, the ligand that initially bound the monomeric protein disappears after the introduction of a PPI (Figure 1C). One representative case is the RAC1 protein from the ER category (PDB ID:1MH1 [20]), whose ligand, the guanine nucleotide, disappears upon binding to the N-terminal DH/PH cassette of Trio in the Trio•Rac1 complex. Another example is actin, an ATPase that is the most abundant protein in eukaryotic cells. In addition to the ATP molecule, this protein was also found to bind to a molecule of lactacystin A, a marine natural product preventing actin from polymerization (PDB ID:1IJJ [22]). The plasma vitamin D-binding protein (DBP) was proven to form a complex with actin as a part of the scavenging system in the plasma for cleaning actin released from damaged cells (PDB ID:1KXP [23]). This complex is included in the OX category in Table 1. Due to the interaction between the DBP and the actin, no ligand binding is found at the original position of the lactacystin A. The ARMSD of the two ATP molecules in both structures is 1.00 Å.

Figure 1D shows a fourth scenario, in which the PPI causes the protein to bind to a new ligand, which is the case for the *E. coli* thioredoxin reductase (TrxR) listed in the ES category in Table 1. In the free state, the reductase contains only one molecule of flavin-adenine dinucleotide (FAD, PDB ID:1CL0 [24]). When TrxR forms a complex with thioredoxin reductase 1 (PDB ID:1F6M [25]), a new molecule of 3-aminopyridine-adenine dinucleotide is introduced into the redox-active disulfide pocket. Both structures contain FAD, and the superposition of the two TrxRs yields an ARMSD of 0.73 Å for the two FADs, suggesting that the addition of a second ligand to the pocket has little effect on the binding between FAD and TrxR.

The fifth and sixth scenarios (Figure 1E,F) involve the fusion and disassembly of the ligand molecules, respectively, after the introduction of PPI. In most cases, this occurs when the ligands are large and complex multimeric compounds, which is usually accompanied by the depolymerization of the multimeric structure and vice versa. Typical examples from the studied dataset are structures with oligomeric poly-N-acetyl-beta-D-glucosamine (poly-NAG) ligands. In these structures, the PPI brought by the second protein might help two neighboring pockets to fuse into one large pocket and the monomer NAG ligand to recombine into one large oligomeric ligand. This was observed in angiotensin-converting enzyme-related carboxypeptidase (Ace2), which bound a tri-NAG near the protein–protein interface when bound to the SARS-coronavirus spike protein (PDB IDs 1R42(A) and 2AJF(A, E) [26,27]). In the opposite case, PPI can reduce the binding pockets. An example of this was observed in the human membrane cofactor protein CD46 when bound to the fiber protein from human adenovirus 21(PDB IDs 1CKL(A) and 3L89(M, A) [28,29]), where an oligomeric ligand disassembled into a monomer NAG ligand.

There were nineteen groups with RMSD values greater than 1Å, of which eleven were in the OG category and four were in the OX category. The disappearance of ligands or the emergence of new ligands caused by PPI did not appear in the OG category and rarely appeared in the ES and the ER. On the contrary, it was more common in the OX. Complex or large ligands are mainly found in OX, AA, and OR; in these groups, PPI frequently leads to more complex ligand changes, including ligand synthesis, decomposition, or disappearance. These data show the diverse effects of PPIs on the protein–ligand interaction, with different degrees of influence on each protein category.

### 2.2. Evaluating the Prediction of mcDPA for Small-Molecule Binding Pockets of Protein–Protein Complexes from the Benchmark

We applied *mc*DPA to predict the ligand-binding pockets in the recently compiled protein–protein complex dataset originally used in developing protein–protein docking algorithms [30]. Table 1 compares the *mc*DPA predictions of the ligand-binding regions with and without protein–protein interactions. For clarity, we excluded complex structures that do not bind to any ligands. We focused more on cases where identical or similar ligands bind to the same constituent protein, both in the apo monomeric protein state or in a state bound to a second protein. As a result, we obtained 63 pairs of small-molecule-protein complex data, of which 52 pairs were non-poly-NAG ligands. The *mc*DPA predicts at least one binding region for each protein *A*, 52 of which (or 82%) overlap with the ligand by at least one atom. Of the remaining eleven cases, six are NAG ligands.

Regarding protein–protein interaction types, the *mc*DPA predictions hit 18 of the 21 members of the OG category. The prediction precision was at least 0.5 for nine and not less than 0.3 for fifteen members. In total, 15 members had predicted recall values of at least 0.5 and 0.3 or more for 17 members. The *mc*DPA hit fourteen of the sixteen members of the OX category; the two remaining members bound to carbohydrate ligands (one was NAG, and the other was BMA). The prediction precision was at least 0.5 for 12 members, with a recall of at least 0.5 for half the members. The ES group had ten members, and the predictions hit nine of them; eight predictions had an accuracy of no less than 0.50, and six predictions had a recall of more than 0.5. The *mc*DPA predictions hit five out of seven members of the ER group, with two predictions having an accuracy of over 0.80 and three predictions having a recall of 0.95 or higher. The *mc*DPA predictions hit three of the five members in the antibody–antigen (AA) category and two of the four members of the OR category. For the AA and OR, while the prediction of the binding pockets for the seven structures that bind NAGs performed poorly, the *mc*DPA predictions hit the non-NAG-binding pockets with significant prediction precision and recall.

Taken together, 33 (52%) of the 63 *mc*DPA-predicted binding regions had a precision of 50% or higher, and 35 cases (55%) had a recall of at least 50%. Comparatively, the *mcDPA* performed better in the ten-member enzyme-substrate (ES) category and the sixteen-member OX class, with accuracies of 0.5 or higher in 80% and 75% of the examined cases, respectively. For these two classes of complexes, the percentages of predictions given by the algorithm with a recall of no less than 0.5 were 60% and 50%, respectively. For the 21-member OG class complex, the algorithm provided an intermediate level of prediction, with 43% of the predictions having an accuracy of no less than 0.5 and 71% having a recall of no less than 0.5. We also note that for a particular class of ligand NAGs in the dataset, the method performs the worst in localizing their binding regions, with an accuracy of less than 10%.

The *mc*DPA prediction of the ligand-binding region was compared to two selected free, standalone programs, FPOCKET 3.0 [31] and CAVIAR 1.1.1 [32] (see Table 2 in reference [17]). The two programs have been applied to 85 structures of the popular CCDC/Astex dataset for ligand-binding site prediction [33]. In total, 65% of FPOCKET 3.0 predictions and 70% of CAVIAR 1.1.1 predictions are at least 30% accurate. As a comparison, the percentage of *mc*DPA predictions on the ZDOCK dataset that achieves this accuracy is 67%, which rises to 78% if structures with ligand NAG are excluded. Although their validation datasets are different, the data volumes are comparable. Thus, our calculations at least suggest that mcDPA is highly competitive in recognizing ligand-binding regions.

### 2.3. Comparison with Binding Pockets of FDA-Approved Small-Molecule Drugs Targeting Protein–Protein Complexes

As an application, we used *mc*DPA to predict binding regions for 33 protein–protein complexes with solved atomic structures. These protein complexes are targets of recently FDA-approved small-molecule drugs, including 15 inhibitors and 18 stabilizers. The algorithm successfully identified the spatial orientation of 29 of the small-molecule drugs. Table 2 compares the binding regions calculated by the *mc*DPA with the binding pockets identified in the solved structure. There are 20 (61%) predictions with a precision of 0.5 or higher and 27 (82%) with a precision higher than 0.30. There are 19 (58%) predictions with a recall of 0.5 or higher (of which 14 have values of 0.80~1.00) and 27 (82%) with a recall higher than 0.30. Compared to the predictions for the structures in the ZDOCK benchmark dataset, the algorithm performs better for protein–protein complex-type targets.

### 2.4. Modeling the Effect of Protein–Protein Interactions on the Prediction of Ligand-Binding Regions

Considering that the addition of PPIs leads to changes in protein conformational dynamics, it was natural to investigate how the *mc*DPA algorithm responds to a given PPI when detecting ligand-binding regions. Appendix A lists the changes in the predicted ligand-binding regions or clusters due to the introduction of a PPI by applying the algorithm to the ZDOCK benchmark dataset. A total of 165 DPA clusters were created by 63 monomeric proteins, of which 73 vanished upon protein–protein interaction. The algorithm generated 268 ligand-binding regions or clusters for the 63 corresponding protein–protein complexes, with 91 overlapping with the clusters derived from the monomeric proteins. The typical changes can be categorized as follows: (1) Twenty-four clusters remained unchanged (shown in the table as O→O′), which accounted for 38% of all the predicted regions, as shown in Figure 2A; (2) eight clusters transferred from different clusters, as shown by P→O′ in Figure 2B; (3) thirty-one clusters were created from empty, indicated in the table by Ø→{O′,P′,Q′}, as shown in Figure 2C; (4) eight clusters vanished, indicated in the table by O→Ø, as shown in Figure 2C; (5) clusters decomposed into two or more clusters, as shown in Figure 2D (O→Q′, O→S′); (6) two or more clusters merged into one cluster, as shown in Figure 2E ({O,P}→Q′). Notably, the predicted changes due to the PPIs were similar to those observed in the experimental ligands determined in the protein–protein-complex structures.

### 2.5. The Effect of Protein–Protein Docking Orientation in the Predicted Binding Regions

Protein–protein contact orientation probabilistically determines the key interactions between two proteins. Thus, it is worthwhile to investigate how the docking orientation between proteins affects the predicted ligand-binding regions. Here, we take the cell-division protein FTSZ and the hypothetical protein PA3008 SULA from *P. aeruginosa* as an example (PDB code:1OFU, 2VAW). The natural ligand for FTSZ/SULA is GDP, and for this complex, the *mc*DPA predicted a binding region with a precision of 0.78 and a recall of 0.86. Now, consider using the popular docking program, ZDOCK 3.0, to generate 100 different protein-complex conformations. Appendix A compares the ligand-binding regions predicted based on different protein–protein orientation conformations. Figure 3 shows the two clusters of docking orientations given by ZDOCK, five typical conformations, and the *mc*DPA prediction regions for each conformation. Similarly to the effect of the PPI on the *mc*DPA prediction, changes in docking orientation can also cause similar types of changes in prediction regions, as follows: a total of 44 clusters keeping O→P′, accounting for 11% of the total predicted regions; 79 clusters vanishing as P/Q→Ø, accounting for 20%; 38 clusters turning to R → Q′, accounting for 10%; 64 clusters keeping O/Q/R → O′, accounting for 16% of the total predictions. These calculations illustrate that small conformational changes due to relative orientation can significantly affect the dynamic conformational distribution of the complexes to varying degrees. The results also show that the *mc*DPA can capture these changes more accurately.

## 3. Materials and Methods

### 3.1. The Test Dataset

To study the effect of PPI on LPI, we needed a protein dataset whose element is involved in both protein–protein interactions and ligand–protein interactions. For this purpose, we used the benchmark 5.5 protein–protein-complex dataset developed by Weng and colleagues [30]. Protein–protein complexes were screened for members that are also involved in known ligand–protein interactions. The original benchmark was a non-redundant dataset developed to test protein–protein docking algorithms, and by 2015, after several updates, benchmark 5.0 had 231 cases. On this basis, we screened candidate proteins that also bind to small-molecule ligands, and, finally, 63 protein datasets were collected. The dataset proteins can be broadly classified into six categories based on their functions, namely ES (the enzyme–substrate type), ER (the enzyme-regulator type), AA (the antibody–antigen type), OG (other G protein-containing type), OR (other receptor-containing types), and OX (other miscellaneous).

### 3.2. Implementing mcDPA

*mcDPA* is a direct extension of the recently developed multilayer dynamics perturbation analysis method (MDPA), which predicts ligand-binding regions for a given monomer protein structure [17,34]. MDPA identifies regions on the protein surface that cause large changes in the distribution of protein conformation using a high-speed perturbation calculation. For a given protein, it first decorates with layers of test points (*s*) on the surface that are randomly distributed and cover the entire protein surface. The algorithm’s key is distinguishing test points near the ligand-binding pocket from other locations on the protein surface. It achieves this by assigning a number, called a dynamic perturbation value, *D*_x_, to each test point and then grouping points with high *D*_x_ values into clusters, which predict the ligand-binding regions. The *D*_x_ value is the Kullback–Leibler divergence, which measures the change in protein conformational distribution due to the interactions of the test points. In practice, the protein conformational distribution near the equilibrium native state is analytically solved using an elastic model. In this model, neighboring atoms are in contact with each other through a spring of force constants *γ* [35,36,37]. The method was tested with the standard single-domain–protein-ligand interaction test set, the CCDC/Astex dataset [33]. For 80% of the test cases, predicted clusters physically overlap with the natural ligands by at least 50%.

Now consider a more complicated case, in which protein *A* binds to a ligand in the presence of a second protein, *B*. In this case, we introduce the *mc*DPA method as a direct extension of MDPA, where an additional elastic network model is introduced to model the interaction between protein *A* and *B*: $V^{AB}=\frac{1}{2}\underset{i,j}{\sum }\gamma^{AB}{\left(r_{ij}-r_{ij}^{0}\right)}^{2}$. Here, $\gamma^{AB}$ represents the force constant of the interaction between the two interface atoms (*i*, *j*) from protein *A* and *B*, respectively. In this way, *A* and *B* are bound together to form a dynamic complex, *B-A*, and its ligand-binding regions can be predicted again by applying the MDPA algorithm to the complex system. Usually, we set $\gamma^{AB}=\gamma$ to indicate that the strengths of atomic interactions at the protein interface are equal to those inside individual proteins. However, $\gamma^{AB}$ is a proper adjustable parameter for studying the effects of PPIs of different strengths on ligand binding.

### 3.3. Structural Characterization of the Effects of PPIs on LPIs

Let *A*-*L* represent the complex formed by the binding of ligand *L* to an individual protein *A*, and *B*-*A*-*L* the complex formed by the binding of the same ligand *L* to the protein–protein complex, *B-A*. The spatial occupancy and orientation of the ligand inside the binding pocket can be represented by the set of ligand atoms $L=\left\{\;l_{i},i=1,2,\ldots ,n_{L}\right\}$, where $n_{L}$ is the total number of ligand atoms. It is then helpful to evaluate the effect of PPI on LPI by measuring changes in the spatial distribution of the ligand *L*. One of the simplest characterization methods is to align the *A* proteins from the two complexes (*A-L* and *B-A-L*, respectively) and then calculate the absolute root mean square derivative (ARMSD) between the *L-*ligands carried by the two complexes (see Figure 4). The larger the ARMSD value, the greater the effect of adding the *B-A* PPI on the LPIs in the *A*-*L*. The *B-A* PPI interactions may perturb the conformation distribution of protein *A*, thus indirectly affecting the binding pocket and the binding of ligand *L* to protein *A*. In other cases, the *B-A* PPI might physically overlap with or be close to the binding pocket, thus directly affecting *A-L* binding.

### 3.4. Computational Characterization of the Effect of PPI on Ligand–Receptor Binding

The predicted ligand-binding region of *mc*DPA is presented as a cluster of protein surface test points, $C=\left\{s_{i},i=1,2,\ldots ,n_{C}\right\}$, where $n_{C}$ is the total number of test points in the cluster. In some cases, *mc*DPA might predict two or more ligand-binding-region clusters. These clusters are sorted according to the averaged dynamic perturbation value of the cluster points, ${\bar{D}}_{X}$, and labeled in descending order as chains *O*, *P*, *Q*, etc., in PDB format. We used superscript to denote prediction clusters, C’, for the binding pockets in the *B-A* protein–protein complex, which are labeled in ${\bar{D}}_{X}$-descending order as chains $O^{\prime }$, $P^{\prime }$, $Q^{\prime }$, etc., in PDB format.

To evaluate the predictions of *mcDPA*, we calculated the overlap between prediction *C* (or C’) and the ligand *L* as follows: $c=\{s_{i}|\;s_{i}\in C,\;d\left(s_{i},L\right)<r_{cutoff}\}$, where $d\left(s_{i},L\right)$ is the shortest distance between the test point $s_{i}$ in *C* and all atoms of ligand *L*, and $r_{cutoff}$ is the effective overlapping cutoff distance, which has a default value of 3.5 Å. The precision of prediction *C* is defined by the ratio $P=\frac{\left|c\right|}{\left|C\right|}$, and the recall of the prediction is defined by the ratio $R=\frac{\left|l\right|}{\left|L\right|}$, where $l=\{l_{i}|\;l_{i}\in L,\;d\left(l_{i},C\right)<r_{cutoff}\}$.

A quantitative comparison of predicted ligand-binding region C made based on monomeric proteins and of $C^{\prime }$ made based on the protein–protein complex is also valuable for evaluating the effects of PPIs on LPIs. The comparison is based on the calculation of the following two point sets that characterize the overlap between C and $C^{\prime }$: $c=\{s_{i}|\;s_{i}\in C,\;d\left(s_{i},C^{\prime }\right)<r_{cutoff}\}$ and $c^{\prime }=\left\{s_{i}|\;s_{i}\in C^{\prime },\;d\left(s_{i},C\right)<r_{cutoff}\right\}$. The two ratios $P=\left|c^{\prime }\right|/\left|C^{\prime }\right|$ and R=c/C measure the difference between $C^{\prime }$ and C, quantifying the impact of PPIs on the LPIs. Before determining the overlap between $C^{\prime }$ and C, the involved *A* proteins from the two complexes (*A-L* and *B-A-L*, respectively) must be aligned, a process similar to that used to determine the AMSD value for ligand.

## 4. Conclusions

In this study, we characterized six typical effects of PPIs on LPIs by examining the various ligand-binding states in 63 pairs of small-molecule–protein complexes from the ZDOCK benchmark dataset. The comparative calculations showed that when PPI is introduced, the ligand may maintain its original binding state or undergo changes, such as migration, disappearance, rebirth, decomposition, and fusion. We developed the mcDPA model by introducing a PPI term into our previously developed model to predict ligand-binding pockets in the protein–protein complexes. The model was validated using the ZDOCK benchmark dataset 5.5 and was found to be comparable to FPOCKET 3.0 and CAVIAR 1.1.1 in finding ligand-binding regions. Notably, the prediction performance of the *mc*DPA is mixed for different protein–protein-complex categories. For example, it has a higher prediction accuracy for the ES category (80% prediction with an accuracy of more than 0.5). In contrast, it has a lower prediction accuracy for AA or OR categories, especially for protein complexes that bind NAG. This reflects the weakness of mcDPA in predicting shallow and wide binding pockets, possibly due to the insensitivity of these regions to dynamic conformational changes. The model performed better when applied to the 33 FDA-approved drugs targeting protein–protein complexes. In this case, 82% of the predictions had both precision and recall of no less than 0.3. Finally, by simulating the effects of the PPIs (including different types of PPIs and different docking postures) on the LPIs, the *mc*DPA predictions showed six typical changes in the predicted binding regions, similar to the experimentally observed ligand-binding changes. However, the detailed relationship between the two is currently outside the scope of this study and may deserve future investigation. The method developed here may help to study the diversity of drug–target interactions in protein–protein complexes and contributes to the design of relevant drugs. The *mc*DPA code is available at http://www.mingbioinfo.online/MCDPA/ (accessed on 17 July 2024).

## Figures and Tables

**Figure 1 ijms-25-09172-f001:**
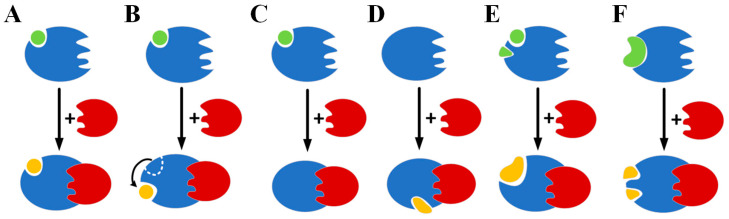
Six typical changes in ligand binding upon introducing protein–protein interactions. Blue for protein A, red for protein B, green for ligand *L*_1_, yellow for ligand *L*_2_. The blue and red represent protein (**A**) Ligand binding is unaffected and remains almost unchanged; (**B**) ligand-binding region shifts; (**C**) ligand disappears; (**D**) a new ligand appears at a site where no ligands were originally bound; (**E**) two neighboring pockets fuse into a big one; (**F**) a binding pocket breaks down into two, usually smaller, binding regions.

**Figure 2 ijms-25-09172-f002:**
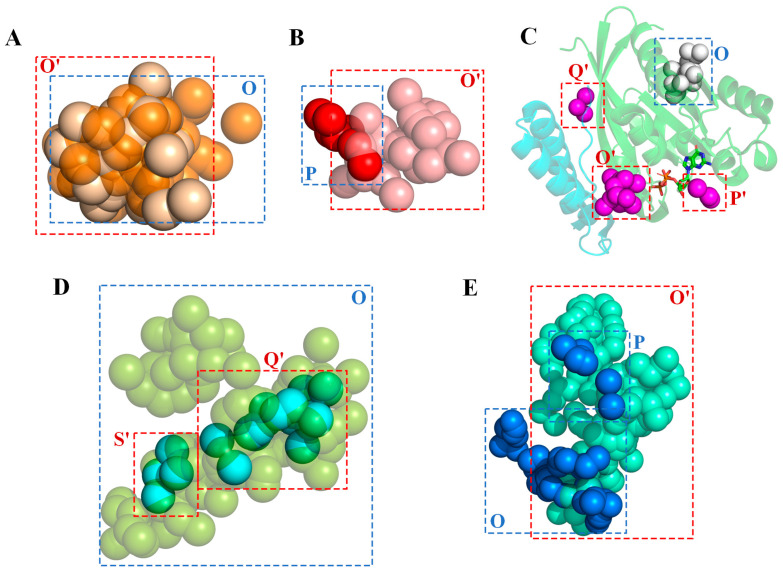
Protein–protein interactions lead to various changes in the predicted ligand-binding regions. (**A**) Clusters remained almost unchanged (Cluster O consists of the orange balls in the blue dashed box and represents the prediction without PPI; Cluster O′ consists of the light brown balls in the red dashed box and represents the prediction with PPI); (**B**) cluster shifting from prediction cluster P→O′; (**C**) cluster disappearance (O**→**Ø) and regeneration (Ø→{O′,P′,Q′**}**); (**D**) cluster disassembly from O→{Q′, S′}; (**E**) cluster merging: {O,P}→Q.

**Figure 3 ijms-25-09172-f003:**
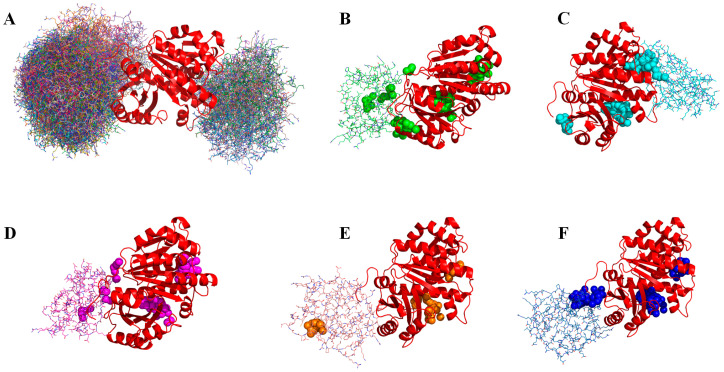
Comparison of mcDPA-predicted regions for 100 conformations generated by ZDOCK. The red cartoon is the cell-division protein FTSZ, and the line structure is the inhibitor protein SULA. Ball sets show small molecule binding regions predicted by mcDPA. (**A**) Two typical classes of docking conformational clusters generated by ZDOCK; (**B**–**F**) five typical docking conformations and their predicted binding regions (clusters of colored balls).

**Figure 4 ijms-25-09172-f004:**
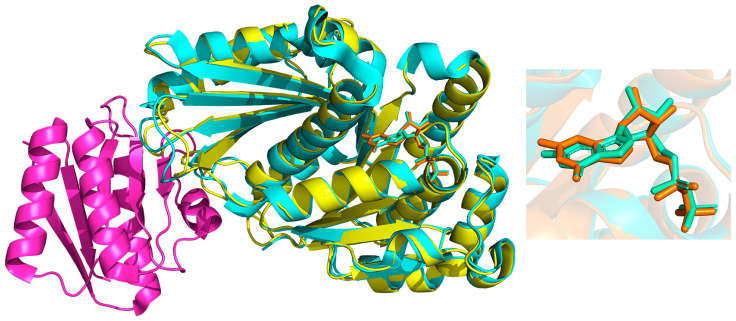
Determining the ligand changes. The protein monomer (Protein A, yellow; Ligand, orange) is superposed with its counterpart (Protein A, cyan; Ligand, green) in the protein–protein complex A + B (cyan and purple). Differences in the atomic distribution of the ligands are shown on the right.

**Table 1 ijms-25-09172-t001:** *mc*DPA predictions of ligand-binding regions in the presence of protein–protein interactions.

Cat.	*A-L* _1_	*L* _1_	*B-A-L* _2_	*L* _2_	ARMSD	*L*_1_ Prediction		*L*_2_ Prediction	
						Precision	Recall	Precision	Recall
OG	1QG4(A)	GDP	1A2K(D,A)	GDP	0.76	0.00	0.00	0.67	0.36
OG	1AZT(A)	GSP	1AZS(C,B)	GSP	0.52	0.27	0.60	0.00	0.00
OG	1MH1(A)	GNP	1E96(A,B)	GTP	1.18	0.43	0.79	0.26	0.45
OG	1TND(A)	GSP	1FQJ(A,B)	GDP	0.45 ^@^	0.00	0.00	0.00	0.00
OG	1GIA(A)	GSP	1GP2(A,B)	GDP	1.56	0.27	0.12	0.13	0.54
OG	1A4R(A)	GDP	1GRN(A,B)	GDP	1.50	0.60	0.79	0.13	0.45
OG	1MH1(A)	GNP	1HE1(C,A)	GDP	1.02	0.43	0.79	0.31	0.94
OG	821P(A)	GNP	1HE8(B,A)	GNP	0.97	0.57	0.70	0.00	0.00
OG	1MH1(A)	GNP	1I4D(D,A)	GDP	1.11	0.43	0.79	0.67	0.38
OG	1QG4(A)	GDP	1IBR(A,B)	GNP	1.47	0.00	0.00	0.00	0.00
OG	1O3Y(A)	GTP	1J2J(A,B)	GTP	0.37	0.82	0.82	0.67	1.00
OG	1RRP(A)	GNP	1K5D(A,B)	GNP	1.32	0.00	0.00	0.00	0.00
OG	5P21(A)	GNP	1LFD(B,A)	GNP	0.99	0.52	0.88	1.00	0.33
OG	1HUR(A)	GDP	1R8S(A,E)	GDP	3.63	0.00	0.00	0.33	1.00
OG	6Q21(A)	GCP	1WQ1(R,G)	GDP	1.41	1.00	0.48	0.00	0.00
OG	2BME(A)	GNP	1Z0K(A,B)	GTP	0.26	0.21	0.75	0.33	0.18
OG	1MH1(A)	GNP	2FJU(A,B)	GSP	0.75	0.43	0.79	0.27	0.36
OG	1Z06(A)	GNP	2G77(B,A)	GDP	1.72	0.86	0.94	0.29	0.21
OG	1GFI(A)	GDP	2GTP(A,D)	GDP	1.52	0.11	0.12	0.00	0.00
OG	1MH1(A)	GNP	2H7V(A,C)	GDP	1.54	0.43	0.79	0.41	0.79
OG	1G16(A)	GDP	3CPH(A,G)	GDP	0.53	0.70	0.69	0.00	0.00
OX	1IJJ(A)	ATP	1ATN(A,D)	ATP	2.18	0.80	0.31	0.00	0.00
OX	3DNI(A)	BMA	1ATN(D,A)	BMA	1.63	0.00	0.00	0.00	0.00
OX	1QRQ(A)	NDP	1EXB(A,E)	NDP	0.49	0.27	0.54	0.44	0.13
OX	1IJJ(A)	ATP	1H1V(A,G)	ATP	1.73	0.80	0.31	0.00	0.00
OX	1KUY(A)	COT	1IB1(E,A)	COT	0.95	0.68	0.89	0.83	0.57
OX	1IJJ(A)	ATP	1KXP(A,D)	ATP	1.79	0.80	0.31	0.00	0.00
OX	1IAM(A)	NAG	1MQ8(A,B)	NAG	1.27	0.00	0.00	0.00	0.00
OX	3MIN(A)	HCA	1N2C(A,B)	HCA	0.62	0.75	0.97	0.71	0.41
OX	2VAW(A)	GDP	1OFU(A,X)	GDP	0.86	0.76	0.96	0.78	0.86
OX	2FXU(A)	ATP	1Y64(A,B)	ATP	1.61	0.75	0.4	0.00	0.00
OX	1IJJ(A)	ATP	2BTF(A,P)	ATP	1.48	0.80	0.31	0.00	0.00
OX	1NG1(A)	GDP	2J7P(A,D)	GNP	2.37	0.35	0.69	0.50	0.58
OX	2IYL(D)	GDP	2J7P(D,A)	GNP	2.42	0.00	0.00	0.50	0.52
OX	3BIX(A)	NAG	3BIW(A,E)	NAG	1.92	0.14	0.75	0.00	0.00
OX	1IJJ(A)	ATP	3DAW(A,B)	ATP	1.64	0.80	0.31	0.00	0.00
OX	3ODQ(A)	HEM	3SZK(D,F)	HEM	1.03	0.77	1.0	0.83	0.91
ES	1E1N(A)	FAD	1E6E(A,B)	FAD	0.62	0.80	0.49	0.00	0.00
ES	1CL0(A)	FAD	1F6M(A,C)	FAD	4.66	0.33	0.09	0.75	0.47
ES	1B39(A)	ATP	1FQ1(B,A)	ATP	3.24	0.37	0.72	0.18	0.94
ES	1XK9(A)	P34	1ZM4(B,A)	TAD	10.27 ^§^	0.43	0.95	1.00	0.67
ES	1U90(A)	GDP	2A9K(A,B)	GDP	1.08	0.00	0.00	0.01	0.07
ES	1J54(A)	TMP	2IDO(A,B)	TMP	1.19	0.59	0.96	0.59	0.74
ES	1CCP(A)	HEM	2PCC(A,B)	HEM	0.25	0.88	0.93	1.00	0.72
ES	1YCC(A)	HEM	2PCC(B,A)	HEM	0.31	0.85	0.78	1.00	0.72
ES	1GIQ(A)	NAD	4H03(A,B)	NAD	2.15	0.92	0.89	0.61	0.82
ES	1IJJ(A)	ATP	4H03(B,A)	ATP	0.64	0.80	0.31	0.00	0.00
ER	1JMJ(A)	NAG	1JMO(A,H)	NDG	5.29	0.00	0.00	0.00	0.00
ER	2CN0(H)	F25	1JMO(H,A)	NAG	24.32 ^‡^	0.82	0.95	0.00	0.00
ER	3C13(A)	EMO	1JWH(A,C)	ANP	6.07 ^§^	0.43	1.00	0.69	0.87
ER	1V8Z(A)	PLP	1WDW(B,A)	PLP	0.33	0.00	0.00	0.00	0.00
ER	1E3T(A)	NAP	2OOR(C,A)	TXP	7.39	0.13	0.12	0.00	0.00
ER	1L7E(A)	NAI	2OOR(A,C)	NAD	1.18	0.28	0.95	0.26	1.00
ER	2YVF(A)	FAD	2YVJ(A,B)	FAD	1.31	0.83	0.25	1.00	0.19
AA	1HRC(A)	HEM	1WEJ(F,H)	HEM	3.42	0.84	0.95	1.00	0.69
AA	1YWH(A)	NAG	2FD6(U,A)	NAG	3.90	0.00	0.00	0.00	0.00
AA	3TGT(A)	NAG	3SE8(G,H)	NAG	1.09	0.00	0.00	0.12	0.11
AA	3TGT(A)	NAG	3U7Y(G,H)	NAG	0.71	0.00	0.00	0.00	0.00
AA	4GT7(A)	NAG	5HYS(G,I)	NAG	1.33	0.14	0.64	0.00	0.00
OR	1JX6(A)	AI2	1ZHH(A,B)	NHE	19.90 ^‡^	0.47	1.00	0.50	0.15
OR	1R42(A)	NAG	2AJF(A,E)	NAG	1.53	0.00	0.00	0.29	0.29
OR	1YWH(A)	NAG	2I9B(E,A)	NAG	5.26	0.00	0.00	0.00	0.00
OR	1CKL(A)	NAG	3L89(M,A)	NAG	4.85	0.00	0.00	0.00	0.00

*A-L*_1_: Complexes of protein monomer (*A*) bound to ligand *L*_1_*. B-A-L*_2_: Complexes in which *A* binds to ligand *L*_2_ in the presence of protein *B* bound to protein *A*. ARMSD: Absolution root mean square deviation of ligand atomic positions. *L*_1_ Prediction: Predictions are based only on the structure of monomer protein *A*. *L*_2_ Prediction: Predictions are based on the structure of protein complex *B*-*A*. ^@^: Small groups that cannot be matched, such as the four-atom PSO_2_ in GSP compared to GDP, are not taken into account in determining ARMSD. ^‡^: As the PPI occupies the ligand-binding pocket, the ligand shifts to a different location, resulting in a large ARMSD. ^§^: The same pocket, with different ligands. Here, ARMSD records the distance between the centers of mass of the two ligands.

**Table 2 ijms-25-09172-t002:** Evaluation of the use of *mc*DPA to predict FDA-approved small-molecule drug binding regions targeting protein–protein complexes.

Activity	Drug	Target PPIs	Target PDB Code	Prediction	
				Precision	Recall
Inhibitor	Navitoclax	BCL2/BAX	4LVT	0.35	0.81
Inhibitor	Venetoclax	BCL2/BAX	6O0K	0.46	1.00
Inhibitor	ABT-737	BCLXUBAK	2YXJ	0.41	0.89
Inhibitor	Maraviroc	CCR5/gp120	4MBS	1.00	0.48
Inhibitor	Tirofiban	FGG/ITGA2B/ITGB3	2VDM	0.34	0.80
Inhibitor	BIO8898	CD40−CD40L	3LKJ	1.00	0.36
Inhibitor	Tacrolimus	FKBP12/CNA/CNB	1BKF	0.81	1.00
Inhibitor	Sirolimus	FKBP12/MTOR	1FAP	0.88	0.98
Inhibitor	Pevonedistat	NEDD8/APPBPI/UBA3	3GZN	0.11	0.42
Inhibitor	AMG-232	P53/MDM2	4OAS	1.00	0.95
Inhibitor	CGM097	P53/MDM2	4ZYF	0.70	0.96
Inhibitor	Nutin-2	P53/MDM2	1RV1	0.83	0.97
Inhibitor	RO-5045337	P53/MDM2	4IPF	0.76	1.00
Inhibitor	SAR-405838	P53/MDM2	5TRF	0.92	0.88
Inhibitor	honokiol	RXR/TIF2	4OC7	0.00	0.00
Stabilizer	Epibestatin	14-3-3/PMA2	3M50	0.41	0.36
Stabilizer	Pyrrolidone1	14-3-3/PMA2	3M51	0.59	0.43
Stabilizer	BMS-202	PD-1L/PD-1L	5J89	0.90	0.73
Stabilizer	BMS-8	PD-1L/PD-1L	5J8O	0.97	0.75
Stabilizer	Compound 3	14-3-3/ChREBP	6YGJ	1.00	0.52
Stabilizer	Fusicoccin	14-3-3/H+-ATPase	2O98	0.49	0.53
Stabilizer	Lenalidomide	CK1α/CRL4	5FQD	0.00	0.00
Stabilizer	CC0651	Cdc34/Ubiquitin 1α	4MDK	0.86	0.49
Stabilizer	GW6471	PPARα/SMRT	1KKQ	0.67	0.80
Stabilizer	2x RO-2443	MDM4/MDM4	3U15	1.00	0.49
Stabilizer	2x RO-2443	MDM2/MDM2	3VBG	1.00	0.49
Stabilizer	2x Tafamidis	TTR/TTR	3TCT	0.00	0.00
Stabilizer	4xTrifluoperazine	S100A4/S100A4	3KO0(A,B)	0.33	0.11
Stabilizer	(R,R)-2a	iGluR2/iGluR2	3BBR	0.00	0.00
Stabilizer	Coumarin	lambda-6A/lambda-6A	6MG5	0.05	0.24
Stabilizer	2x NS309	CaM/CaMBD2-a	4J9Z	1.00	0.64
Stabilizer	Inositol tetraphosphate	HDAC3/SMRT	4A69	0.85	0.92
Stabilizer	FK506	FKBP12/calcineurin	1TCO	0.67	0.93

## Data Availability

Data are contained within the article and Appendix A.

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
