# Peer review of "Quantitative Characterization of the Impact of Protein–Protein Interactions on Ligand–Protein Binding: A Multi-Chain Dynamics Perturbation Analysis Method"

_ijms, 2024, doi:10.3390/ijms25179172_

Round 1
Reviewer 1 Report
Comments and Suggestions for Authors
Based on a thorough review, I recommend rejecting the paper titled "Quantitative characterization of protein-protein interactions on ligand-protein binding: a multi-chain dynamics perturbation analysis method" for the following reasons:
1. The title and abstract play a crucial role in guiding the reader's understanding of the study. It's essential that they accurately communicate the focus on computational methods. A lack of clarity in this area can mislead readers about the nature of the research and its contributions. Additionally, one section is described 2 times. please check the introduction.
2. Complex Language: The language used is overly complex and dense, making it difficult for readers to understand the key points of the study quickly. Simplifying the language and breaking down complex sentences significantly improve readability.
3. All abbreviations should be clearly defined upon first use and used consistently throughout the paper.
4. Insufficient Detail: The methodology section needs more detail about the experimental design and the rationale behind the choice of the dataset. Readers need a clearer understanding of why specific datasets and computational models were chosen. The explanation of the multi-chain dynamics perturbation analysis (mcDPA) model is overly technical and not accessible to readers who are not already familiar with the underlying concepts. A more intuitive explanation or graphical representation would be beneficial.
5. The results section provides extensive numerical data without sufficient interpretation or context. The significance of these results in the broader context of protein-protein and ligand-protein interactions is not adequately discussed. The discussion section fails to explore the implications of the findings in depth. It does not adequately address the limitations of the study or provide a balanced view of the strengths and weaknesses of the mcDPA model.
6. The figures and tables need to be well-described, and their relevance needs to be clearly explained. Detailed captions and better integration into the text are necessary to make these visual aids more effective.
7. The conclusion needs to critically analyze the performance of the mcDPA model across different protein-protein complex categories and address areas where the model performs poorly.
Given these significant issues, the paper in its current form does not meet the standards for publication. A significant revision addressing these points is necessary before reconsideration for publication.
Comments on the Quality of English LanguageComplex Language: The language used is overly complex and dense, making it difficult for readers to understand the key points of the study quickly. Simplifying the language and breaking down complex sentences significantly improve readability.
Author Response
Dear Editor,
We greatly appreciate reviewer 1's comments. We have carefully read all the comments and suggestions and have carefully revised the manuscript accordingly in light of these comments. We have itemized our responses in red below.
Best Regards,
Ming
Based on a thorough review, I recommend rejecting the paper titled "Quantitative characterization of protein-protein interactions on ligand-protein binding: a multi-chain dynamics perturbation analysis method" for the following reasons:
- The title and abstract play a crucial role in guiding the reader's understanding of the study. It's essential that they accurately communicate the focus on computational methods. A lack of clarity in this area can mislead readers about the nature of the research and its contributions.
We slightly revised the title by emphasizing the effect of PPI on LPI.
As far as we know, this study is the first systematic discussion of this phenomenon. Therefore, statistical analysis based on databases and developing model prediction methods are equally important. This is why our title does not focus exclusively on methodological research.
Additionally, one section is described 2 times. please check the introduction.
Thanks to the reviewer for pointing this out; we have removed the duplicate paragraph.
- Complex Language: The language used is overly complex and dense, making it difficult for readers to understand the key points of the study quickly. Simplifying the language and breaking down complex sentences significantly improve readability.
This is a good suggestion!We have sought the help of native-speakers and have tried to revise the whole text throughout to simplify the presentation, especially to break down complex sentences.
- All abbreviations should be clearly defined upon first use and used consistently throughout the paper.
Great advice. We've gone through a manuscript from scratch and checked all the abbreviations.
- Insufficient Detail: The methodology section needs more detail about the experimental design and the rationale behind the choice of the dataset. Readers need a clearer understanding of why specific datasets and computational models were chosen. The explanation of the multi-chain dynamics perturbation analysis (mcDPA) model is overly technical and not accessible to readers who are not already familiar with the underlying concepts. A more intuitive explanation or graphical representation would be beneficial.
As we mentioned in the METHOD section (line 635), mcDPA is an extension of our recently developed model MDPA by adding a protein-protein interaction term (line 662). It evolved from DPA à FDPA à MDPA à mcDPA. The basic idea has been summarized in lines 635~655, and the plots and equations involving the algorithm can be found in cited papers. In addition, the code we provide to implement the algorithm (Fortran + Perl) contains some instructions.
- The results section provides extensive numerical data without sufficient interpretation or context. The significance of these results in the broader context of protein-protein and ligand-protein interactions is not adequately discussed. The discussion section fails to explore the implications of the findings in depth. It does not adequately address the limitations of the study or provide a balanced view of the strengths and weaknesses of the mcDPA model.
We have added a separate paragraph with some comparisons to literature methods such as FPOCKET/CAVIAR (line 500 ~508). Since the method evolved from its predecessor DPA, it also has similar weaknesses, such as the inability to predict binding pockets at fully closed voids.
- The figures and tables need to be well-described, and their relevance needs to be clearly explained. Detailed captions and better integration into the text are necessary to make these visual aids more effective.
Good suggestion. We have added a more detailed description to the figures and tables.
- The conclusion needs to critically analyze the performance of the mcDPA model across different protein-protein complex categories and address areas where the model performs poorly.
We revised the conclusion, mentioning the model's weaknesses and strengths and subsequent issues to be addressed.
Given these significant issues, the paper in its current form does not meet the standards for publication. A significant revision addressing these points is necessary before reconsideration for publication.
We believe the topic discussed in this manuscript should help draw academic attention to the important problem of the interaction between PPIs and LPIs. Our model provides a useful attempt to address this issue.
Reviewer 2 Report
Comments and Suggestions for Authors
In this manuscript Li et al conducted a detailed evaluation of protein-protein interactions, and the binding pockets for the various compounds. This article is very interesting and can be published.
One suggestion that it will be better to plot the binding affinity or the binding energy of the protein studied.
Author Response
Dear Editor,
We greatly appreciate reviewer 1's comments. We have carefully read all the comments and suggestions and have carefully revised the manuscript accordingly in light of these comments. We have itemized our responses in red below.
Best Regards,
Ming
=====================================================================
In this manuscript Li et al conducted a detailed evaluation of protein-protein interactions, and the binding pockets for the various compounds. This article is very interesting and can be published.
We are grateful to the reviewers for their comments.
One suggestion that it will be better to plot the binding affinity or the binding energy of the protein studied.
We agree that binding affinity/energy is key in determining ligand-protein interactions, including ligand-protein-protein complex interactions. Unfortunately, the simplicity of our current method does not allow us to give a numerical value for the binding energy, which needs to be obtained with the help of other tools, such as the vina dock/rosetta dock, and obtaining an accurate binding energy is still a very challenging topic. The perturbation value Dx, which we have given, reflects a characteristic quantity of the protein itself and is unrelated to the specific ligand compound. Therefore, it does not reflect the binding energy.
Indeed, this study aims to find the location and shape of potential ligand binding pockets on the surface of proteins. The review's suggestion makes us wonder if it is possible to find a model that simultaneously calculates binding pockets, binding regions, and binding energies. Although this is beyond the scope of this paper, it may be worth a follow-up study.
Round 2
Reviewer 1 Report
Comments and Suggestions for Authors
The authors had improved the paper, explaining in details my concerns.
i suggest to accept this new version